# Future movement plans interact in sequential arm movements

Mehrdad Kashefi[1], Sasha Reschechtko[2,3], Giacomo Ariani[1,4], Mahdiyar Shahbazi[1], Alice Tan[1], Jörn Diedrichsen[1,4,5], J Andrew Pruszynski[1,3]*

[1]Western Institute for Neuroscience, Western University, London, Canada; [2]School of Exercise and Nutritional Sciences, San Diego State University, San Diego, United States; [3]Department of Physiology and Pharmacology, Western University, London, Canada; [4]Department of Computer Science, Western University, London, Canada; [5]Department of Statistical and Actuarial Sciences, Western University, London, Canada

**Abstract** Real-world actions often comprise a series of movements that cannot be entirely planned before initiation. When these actions are executed rapidly, the planning of multiple future movements needs to occur simultaneously with the ongoing action. How the brain solves this task remains unknown. Here, we address this question with a new sequential arm reaching paradigm that manipulates how many future reaches are available for planning while controlling execution of the ongoing reach. We show that participants plan at least two future reaches simultaneously with an ongoing reach. Further, the planning processes of the two future reaches are not independent of one another. Evidence that the planning processes interact is twofold. First, correcting for a visual perturbation of the ongoing reach target is slower when more future reaches are planned. Second, the curvature of the current reach is modified based on the next reach only when their planning processes temporally overlap. These interactions between future planning processes may enable smooth production of sequential actions by linking individual segments of a long sequence at the level of motor planning.

**\*For correspondence:**
andrew.pruszynski@uwo.ca

## eLife assessment

This study presents an **important** set of results illuminating how movement sequences are planned. Using several different behavioural manipulations and analysis methods, the authors present **compelling** evidence that multiple future movements are planned simultaneously with execution, and that these future movement plans influence each other. The work will be of great interest to those studying motor control.

## Introduction

Many everyday actions like speaking or preparing a cup of tea are composed of a long and often rapid sequences of movements (*Lashley, 1951*). For successful performance of such tasks, the next movement needs to be proactively planned before the previous movement is concluded. Indeed, prior investigations in saccadic eye movements (*McPeek et al., 2000*; *McPeek and Keller, 2002*), reading (*Rayner, 1998*), walking (*Patla and Vickers, 2003*), typing (*Snyder and Logan, 2014*), finger movements (*Ariani et al., 2021*; *Ariani et al., 2020*; *Shahbazi et al., 2024*), path tracking (*Bashford et al., 2022*), target harvesting (*Diamond et al., 2017*), and reaching (*Howard et al., 2015*; *Säfström et al., 2014*; *Zimnik and Churchland, 2021*) consistently show that sequence production is faster and more efficient when participants have access to information that allows them to plan the future

movements. This improvement demonstrates the nervous system's ability to plan future movements while executing the current movement – i.e., to do *online planning* (*Ariani et al., 2021*; *Ariani et al., 2020*; *Ariani and Diedrichsen, 2019*).

Planning and execution-related processes of a single movement occur in overlapping brain areas and often even carried out by the same neurons (*Crammond and Kalaska, 2000*; *Elsayed et al., 2016*; *Kaufman et al., 2014*), so an important question is how the nervous system avoids interference between the planning of a future movement and the control of the current one when producing rapid sequential movements. In a short sequence of two reaches, *Zimnik and Churchland, 2021* proposed that in monkey primary motor cortex (M1) and dorsal premotor cortex (PMd), preparation of the next movement occurs in an orthogonal neural subspace to that which controls the ongoing movement, thereby allowing these two processes to run in parallel without interference.

For longer movement sequences, especially if they are to be rapidly executed, it may be necessary to prepare beyond the next reach. It remains unknown to what degree multiple future reaches are planned, and whether these planning processes interact with each other and with the ongoing action. Here, we address this question with a new continuous reaching task in which we control how many future movements can be planned and how much the kinematics of the individual segments of the sequence could affect each other.

## Results

We investigated how multiple future targets of a sequence are planned in a continuous reaching task. Participants were instructed to perform sequences of 14 reaches in a planar robotic exoskeleton. The targets were generated from a hexagonal grid of potential targets with radii of 1 cm spaced 4 cm apart over a 21×24 cm² total workspace (*Figure 1A*). Every trial started from the same 'home' target in the center of the workspace. Participants were instructed to capture a target before moving on the next target. They captured each target by staying within it for 75, 200, or 400 ms (dwell time, *Figure 1B*). Longer dwell times required a full stop in each target, while shorter dwell times allowed participants to link subsequent reaches into a co-articulated unit (*Figure 1C*).

Participants could see the position of their hand displayed as a circular cursor in the horizontal plane of the task. Participants were shown either one (Horizon 1), two (H2), three (H3), four (H4), or five (H5) future targets to control how much information about the future sequence was available. The order of future targets was indicated by their brightness. The Horizon 1 condition was equivalent to a serial reaction time task because the next target appeared only when the current one was captured. Therefore, the next movement could not be planned until the end of the current movement (*Figure 1D*, H1). In contrast, the Horizon 2 condition allowed for some planning of the next movement while executing the current one (*Figure 1D*, H2). Horizon 3–5 conditions allowed planning the next two, three, or four movements, respectively (*Figure 1D*, H3).

### Planning future reaches speeds up sequence execution

To establish how many future movements participants planned, we first asked whether participants were faster when extra future targets were visible. To quantify speed, we measured the inter-reach interval (IRI), defined as the time required to move the hand from the boundary of one target to the boundary of the next target (*Figure 2*). IRI was significantly reduced from Horizon 1 to Horizon 2 for all dwell times. The average reduction of IRI was 206 ms ($t_{(10)}$ = 22.76, p=3.02e-10), 232 ms ($t_{(10)}$ = 27.41, p=4.83e-11), and 246 ms ($t_{(10)}$ = 24.84, p=1.27e-10) for the 75, 200, and 400 ms dwell times, respectively. We also observed a further small 16 ms improvement from H2 to H3 in the 75 ms dwell time condition ($t_{(10)}$ = 3.137, p=5.30e-3). These results suggest that, at least for a dwell time of 75 ms, participants plan two targets ahead of the current reach.

### Target jump confirms participants plan two future reaches

Because the reduction of IRI from H2 to H3 observed above was small, we performed a second experiment to test whether participants planned two movements into the future. That is, we occasionally displaced the target two reaches in the future (i.e. the +2 target) when the current (i.e. +0) target was captured. If information about the +2 target was not being used, we would expect to see no interruption in the sequence: both the movement toward the unperturbed +1 target, as well as to the

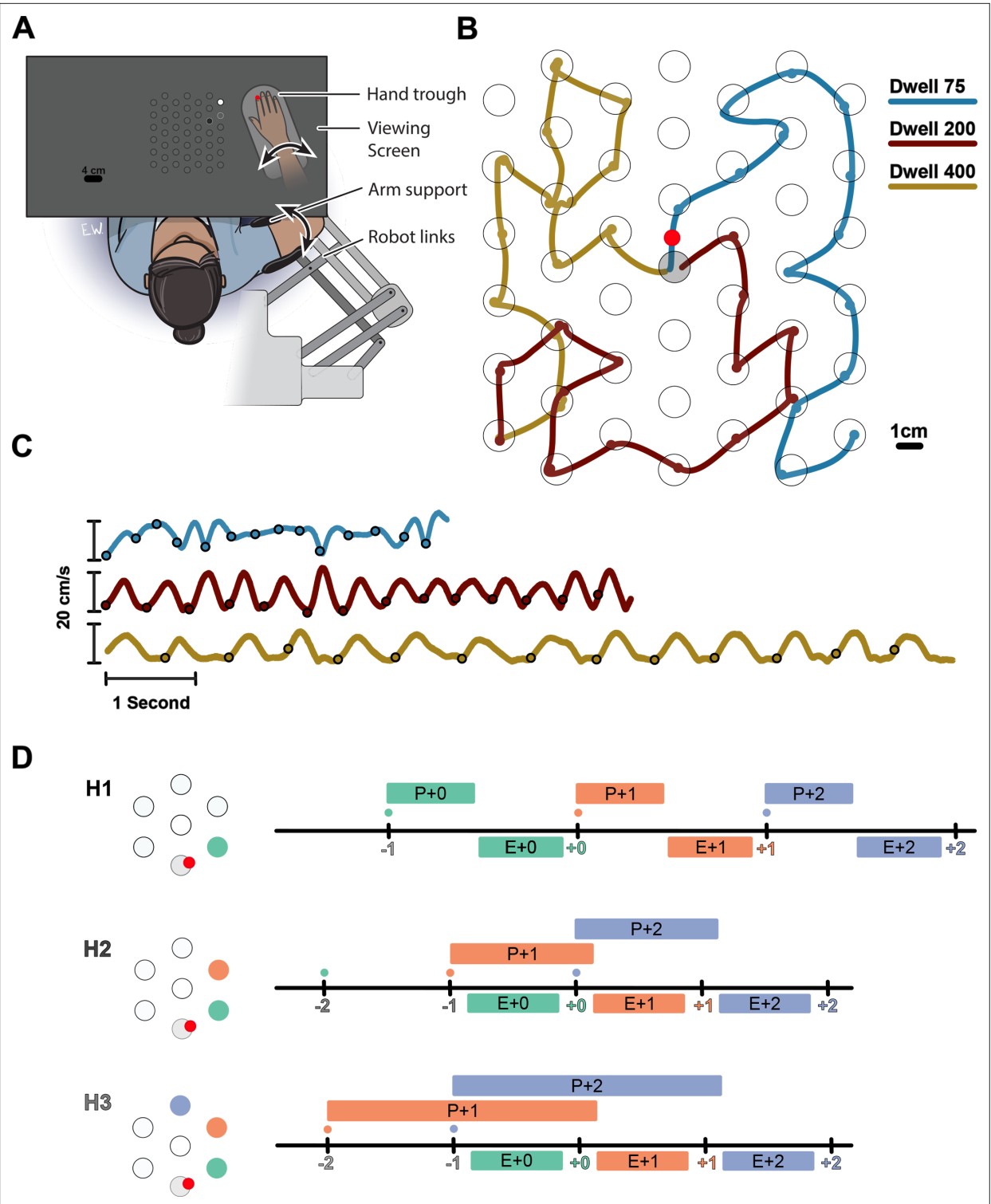

**Figure 1.** Experimental paradigm. (**A**) Participants performed reaches in an exoskeleton robot. Their hand was occluded, and hand position was indicated by a red dot. The full grid of possible targets not shown to participants. The targets and their order were shown with decreasing brightness (an H3 trial is shown). (**B**) Movement trajectory in three example trials (Horizon 2; Dwell 75, 200, 400). Trials always started from a fixed home target in center (gray target). The small circles on the traces show the time point in which the target was captured. (**C**) Speed profiles for the example trials shown in (**B**). (**D**) Timeline of the task for Horizon 1–3 conditions. Ticks show the time when the target was captured (colored number) and a new target was shown on the screen (colored small dot). The boxes above the line show the available time for planning each movement, the time from when the target first shown

*Figure 1 continued on next page*

*Figure 1 continued*

to the beginning of the execution of the movement. The boxes below the line show the execution of each movement, the time interval in which the hand was moving from one target to another.

The online version of this article includes the following figure supplement(s) for figure 1:

**Figure supplement 1.** Speed profile for two participants.

jumped +2 target should not differ from unperturbed conditions (*Figure 3B*, solid line). We tested this prediction in the H3 condition with 75 ms dwell time.

Our results indicate that participants used the information about the +2 target. We observed a normal reach toward the unperturbed +1 target. The reach time to the +1 target was not reliably different between the jump and no-jump conditions ($t_{(9)}$ = 0.63, p=0.54; *Figure 3C*, Execution +1). This was also true for dwell time inside the +1 target ($t_{(9)}$ = 1.98, p=0.08; *Figure 3C*, Dwell +1). However, movement time from the +1 to the +2 target was significantly longer in the jump condition ($t_{(9)}$=5.90, p=2.00e-4; *Figure 3C*, Execution +2).

One reason for this delay could be that visual displacement of the target was simply a distracting stimulus. However, this explanation is not consistent with our kinematic analysis which revealed participants reached toward the pre-jump +2 target, and then corrected their reach toward the new position of +2 target (*Figure 3B*, dotted line). We quantified this commitment to the pre-jump +2 target position by measuring the minimum distance between the reach trajectory and the center of the pre-jump +2 target (see Methods). The minimum distance was significantly lower in the jump condition ($t_{(9)}$=5.78, p=3.00e-4).

Together with the speedup of the overall movement (*Figure 2*), these results show that the reach to the +2 target was at least partially planned before the target jump, simultaneous with the reach to the +0 target and planning of the reach to the +1 target (*Figure 3A*).

## Planning processes for multiple future movements are not independent

Our previously described results indicate that multiple future movements are planned at the same time. Consequently, we next asked whether these preparatory processes are independent of each other or if they interact.

We tested whether two future movement plans interact by jumping the position of the +1 target when the +0 target was captured. This was done under the 75 ms dwell time. We compared the speed of the correction in the H2 and H3 conditions (see *Figure 4A* and Methods). Although the two conditions had similar kinematics, participants could only plan the +1 target in the H2 condition, whereas they could plan both the +1 and +2 targets in the H3 condition (*Figure 4A*). If the +1 and +2 targets are planned independently, the movement correction to a displacement of the +1 target should be the same in the H2 and H3 conditions. Alternatively, if the movements interact – if they are planned

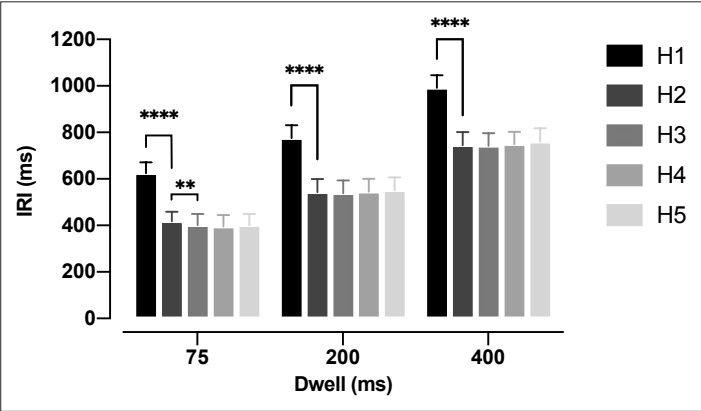

**Figure 2.** Inter-reach interval (IRI) for three dwell times and five horizons. IRI was averaged across all trials, all session, for each participant. The error bars show a 95% confidence interval accross participants (n = 11), ** signifies p<0.01, **** signifies p<0.0001.

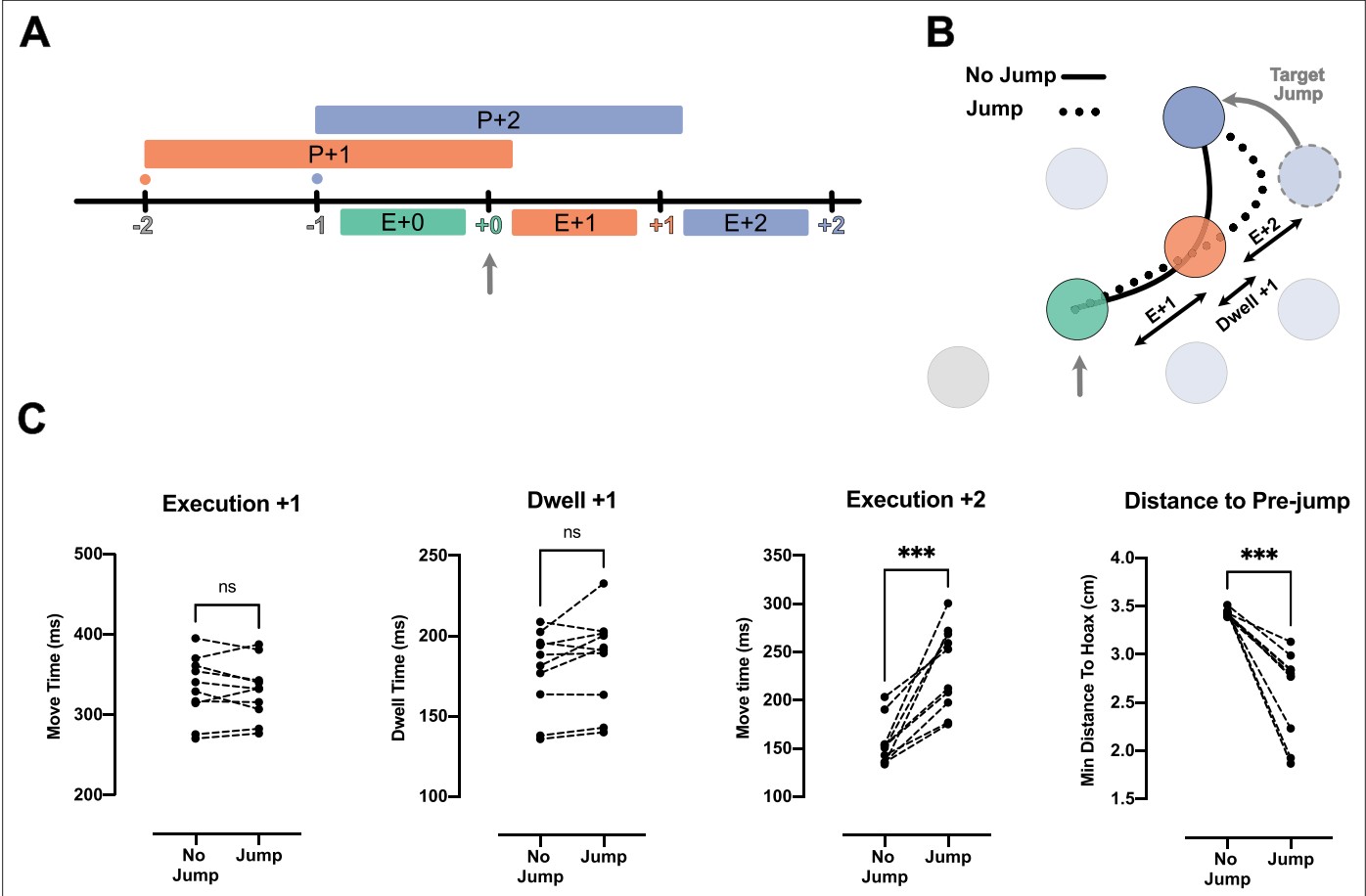

**Figure 3.** Jump of the +2 target reveals existence of planning processes for the reach toward +2 target during the execution of +0 reach. (**A**) Timeline of the jump experiment in Horizon 3, Dwell time 75 ms condition. The jump occurred at the capture of +0 target (vertical gray arrow). (**B**) Reach trajectory for an example no-jump trial (solid line) in which the pre-jump target (light purple) was not shown, and a +2 jump sample trial (dotted line) in which the pre-jump target moved to a new position (dark purple) at the time the +0 target was captured (vertical gray arrow). (**C**) The time for Execution +1 (E+1), Dwell +1, and Execution +2 (E+2), and the minimum distance of reach trajectory to the center of pre-jump target for no-jump and jump conditions. Each dot represents one participant (n = 10), *** shows p-value<0.001.

together or share limited resources – the correction should be slower in the H3 condition because some of the resources would be assigned to planning the +2 target.

Consistent with an interaction between future plans, we found that the corrections for a +1 target jump were longer and slower in the H3 condition than in the H2 condition (*Figure 4B*). In both conditions, participants failed to correct the movement before arriving at the pre-jump position of the +1 target (*Figure 4B*). In the H3 condition, both the movement time ($t_{(9)}$ = 4.85, p=1.80e-3) and the trajectory length ($t_{(9)}$ = 6.19, p=3.00e-4) of the corrective movement were longer than that of the H2 condition. The longer correction trajectory was due to participants moving onward to the +2 target without having corrected for the displaced +1 target. We again used the minimum distance between the corrective reach trajectory and the +2 target to quantify this effect (*Figure 4C*). The corrective reaches were closer to the center of the +2 target in H3 condition ($t_{(9)}$ = 4.28, p=4.00e-3).

In summary, it took more time to update the +1 movement plan when participants could simultaneously plan both the +1 and +2 target as compared to when they could only plan the +1 target. This effect indicates that planning a reach to the +2 target occupied some part of a shared computational resource such that less of the resource was available for updating the reach to the +1 target. These results indicate a clear interaction between the planning processes for future movements.

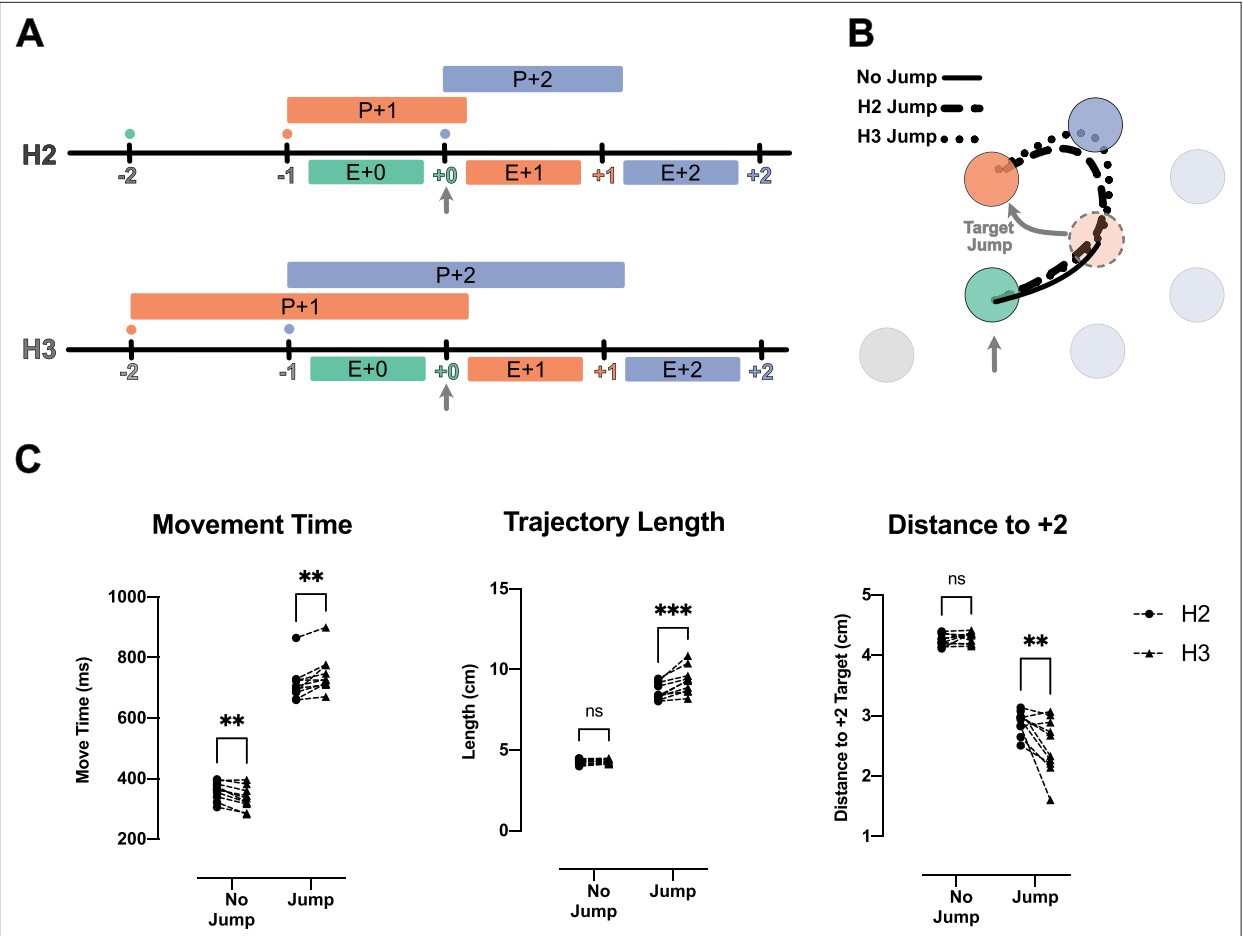

**Figure 4.** Correction for jump of the +1 target is delayed when more future movements are planned. (**A**) Timeline of the jump experiment in the H2 and H3 conditions. In both conditions, the jump of the +1 target (orange) occurred when the +0 target was captured (vertical gray arrow). (**B**) Example trials for a no-jump condition (solid line) and for jump conditions for H2 (dashed line) and H3 (dotted line). In the latter two conditions, the +1 target (orange dotted circle) jumped to a new position (curved gray arrow), when the +0 target was captured (vertical gray arrow). (**C**) Movement time, trajectory length, and minimum distance of the trajectory to the center of +2 target for the reach to the new position of the +1 target. Dots and triangles show mean values for each subject (n = 10) in H2 and H3 conditions respectively. ** and *** signify p-value<0.01 and p-value<0.001, respectively.

### Planning processes are not completely integrated in a single chunk

So far, we have established that people plan reaches to multiple future targets and that these planning processes interact with each other (*Figure 5A*). An extreme version of such an interaction is that the two future reaches are planned as a single unit, as a 'motor chunk' (*Figure 5B*; *Ramkumar et al., 2016*). Our data, however, are not consistent with the idea that future reaches are planned as a motor chunk.

First, an important indicator of chunking is that, after a chunk is executed, there is a short delay until the next chunk is planned. For instance, in the H2 condition, since two targets are shown on the screen at any given time, the participants could execute two fast reaches, followed by a long pause in which they prepared the next chunk of two reaches. However, except for H1 condition, where the participants had to pause and wait for the next target to appear, the speed profiles of other horizon conditions showed no evidence of such pauses (*Figure 1—figure supplement 1*).

Second, chunked planning predicts that a disturbance in any segment of the chunk would affect the whole chunked segment. We tested this prediction in two types of jump experiments. In the H3 condition, where participants could potentially plan +1 and +2 reaches as one chunk, we occasionally either jumped the position of the +2 target (*Figure 5C*) or the +1 targets (*Figure 5D*) when the +0 target was captured. Chunked planning predicts that both jumps should cause a disturbance in the first reach of the chunk, but this did not occur. When we jumped the +2 target, the participants

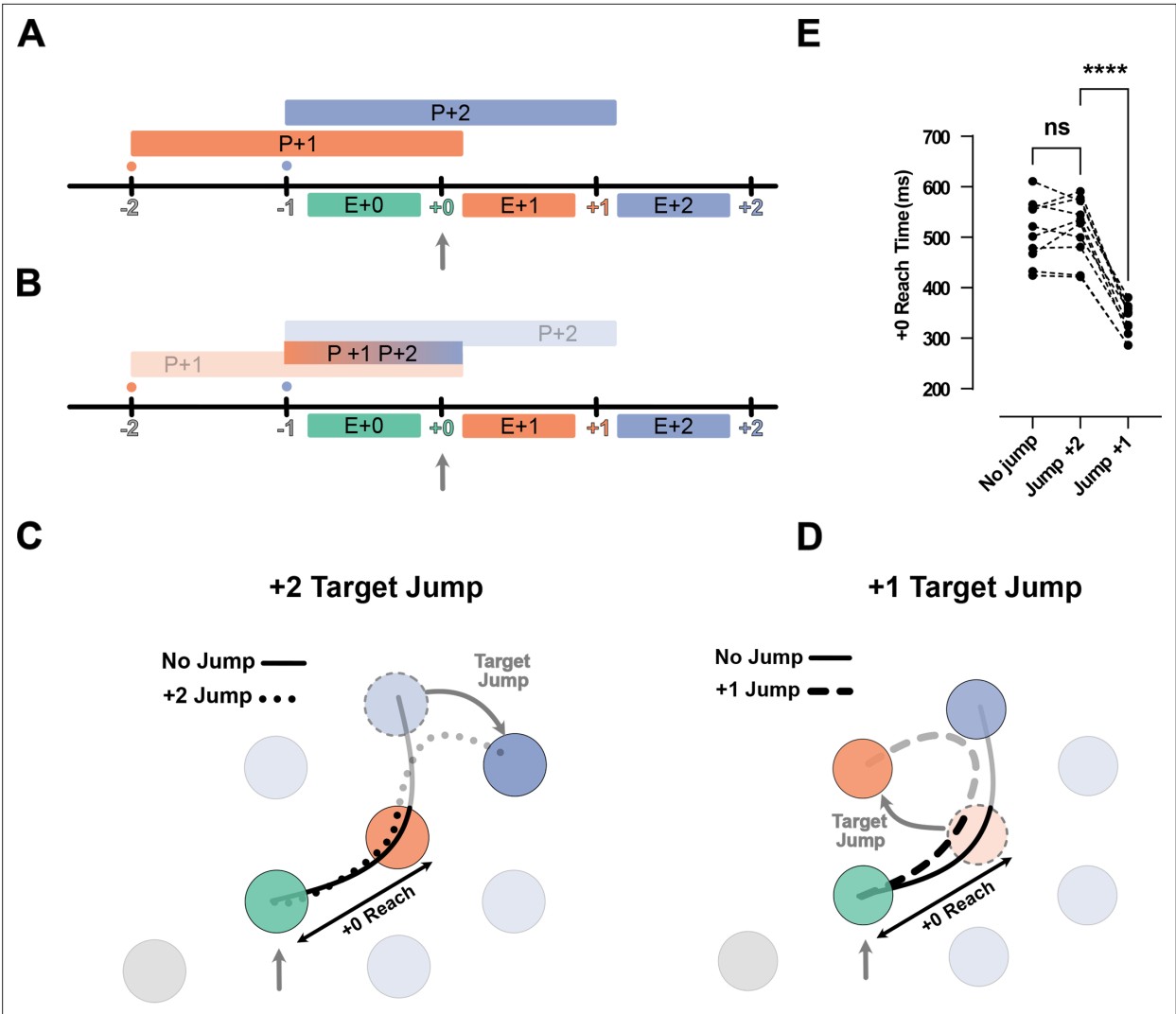

**Figure 5.** Jump of the +2 target or +1 rejects chunked planning of the reaches to the +1 and +2 targets. (**A**) Timeline of the task in the jump experiments. Ticks show the time when the target was captured (colored number) and a new target was shown on the screen (colored small dot). The boxes above the line show the available time for planning each movement, and the boxes below the line show the execution of each planned movement. (**B**) Same as (**A**), but for the chunked planning hypothesis, here one chunked planning controls both Execution +1 (E+1) and Execution +2 (E+2) reaches. (**C**) Reach trajectories for +2 target jump experiment. Reach trajectory for one example no-jump trial (solid line) in which the pre-jump target (light purple) was not shown, and a +2 jump sample trial (dotted line) in which the pre-jump target moved to a new position (dark purple) at the time the +0 target was captured (vertical gray arrow). (**D**) Example trials for a no-jump condition (solid line) and for jump conditions for H2 (dashed line) in which the +1 target (orange dotted circle) jumped to a new position (curved gray arrow) when the +0 target was captured (vertical gray arrow). (**E**) +0 Reach time measured from when the cursor entered the +0 target (green circle) to when it exits the +1 target (orange circle) for each participant (n = 10). *** signify p-value<0.01 and p-value<0.001.

performed the reach to the +1 target in a manner identical to the no-jump condition ($t_{(9)}$ = 0.60, p=0.56). Additionally, when we jumped the +1 target they still went through the pre-jump +1 target but their movement was significantly shortened by deviating toward the new position of the +1 target ($t_{(9)}$ = 14.36, p=1.64e-7) (*Figure 5E*).

In summary, participants were able to correct the second segment of their movement without causing any disturbance in the first segment. This observation provides clear evidence against chucked planning of the future reaches.

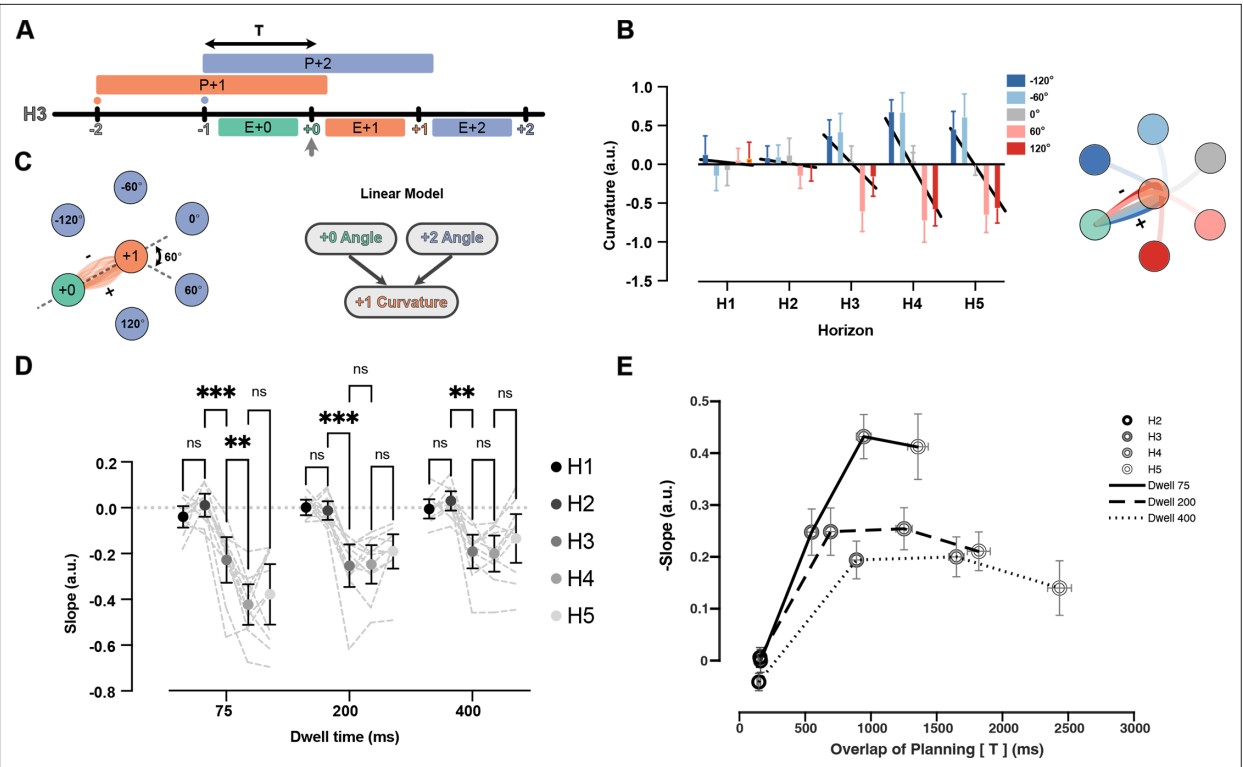

**Figure 6.** Curvature of the reach to +1 target is modulated by position of +2 target. (**A**) Timeline of planning and execution for a Horizon 3 trial. Amount of time overlap (T) between planning processes for reach to +1 and +2 target is represented by the black arrow. (**B**) Effect of +2 target angle on the curvature of +1 reach for Dwell time 75, all horizons, one participant. Positive value of curvature indicates downward curve and vice versa. The overall effect is captured by the slope of the line relating average curvature to the five angles (slope of black lines). (**C**) All the reaches are aligned to one start point and one direction. Then, the angle at the start of the movement to the +2 target can be −120, –60, 0, 60, 120 degrees (60 is shown with dotted line). A linear model is used to predict the signed curvature based on the position of last target (+0 angle) and the +2 target (+2 angle). (**D**) Each dot represents the average summary statistics of the curvature across participants. Individual participant values are shown with shadowed dotted lines in the background. (**E**) Average curvature effect across participants vs overlaps of planning time (T) for each condition. Shades of gray show different horizons and solid, dashed, and dotted lines represent different dwell time conditions. Error bars are SEM across participants (n = 11), and ** signifies p-value<0.001.

## Interaction among planning processes leads to co-articulation of reach segments

The experiments above indicate that reach planning to the +1 and +2 targets interact with each other. Such interactions could allow the motor system to optimize the set of movements leading to systematic co-articulation of movement segments. In other words, when the visual information of the future target is available, each movement in the sequence could be planned in a way that accounts for the movement that comes after it.

Indeed, we observed systematic co-articulation of movements in the H3 condition (*Figure 6A*). When the +2 target demanded an upcoming rightward turn, the +1 reach curved left, and vice versa (*Figure 6B*). Although this deviation led to a longer overall trajectory, it reduced the required turning angle at the +1 target. To summarize the effect of future target on curvature, we fit a linear model that predicted the signed curvature value of the current reach based on two independent variables: the turning angle toward the +2 target, and the incoming angle of the previous reach (see *Figure 6C* and Methods). Note that this model has the advantage over simple averaging because it accounts for the trivial curvature changes caused by the previous movement. The model was fit for each dwell time and horizon separately. *Figure 6C* shows the average curvature for all possible +1 target angles, corrected for the influence of the last target. To summarize the co-articulation effect across all the angles, we fit a line between five values of angles and the curvature (*Figure 6B*, black line). The slope of the line summarizes the strength of the curvature effect (*Figure 6D*) for each dwell time and horizon condition.

In the H1 and H2 conditions, the slope was not reliably different from zero, indicating no systematic co-articulation. This observation is expected for H1 since the +2 target was not on the screen during the planning or execution of the investigated reach. Notably, we did not observe the curvature effect in the H2 condition in which the +2 target was on the screen only during the execution of the +1 reach. This suggests that co-articulation of segments cannot happen if planning of the next segment happens during the execution of the previous one. In the H3 condition, the slope was reliably smaller than zero for all the dwell time conditions, indicating systematic co-articulation once parallel planning was feasible. In the case of the 75 ms dwell time, the co-articulation kept growing from H3 to H4 ($t_{(10)}$ = 5.54, p=2.60e-03). We observed no reliable increase in co-articulation for Horizon>3 for the 200 ms ($t_{(10)}$ = 0.19, p>0.98) or 400 ms dwell times ($t_{(10)}$ = 0.38, p>0.99).

Overall, we observed less co-articulation for longer dwell times. Dwell time can have a dual role here. On the one hand, longer dwell times mean the participants have more time to benefit from future targets because they see the targets longer, potentially leading to more co-articulation. On the other hand, longer dwell times mean the participants had to stay stationary in the target for longer, making the movements less mechanically integrated, and therefore decreasing the benefit of co-articulation. To distinguish between these potential contributions of dwell time, we plotted the curvature effect versus the time that participants could see both the +1 and +2 targets before starting the +1 movement (*Figure 6E*). Both horizon and dwell time led to more overlap in planning times. For all dwell time conditions, the curvature effect increased between H2 and H3 conditions, and then saturated after the H4 condition. However, with longer dwell times, the overall rise and saturation of the

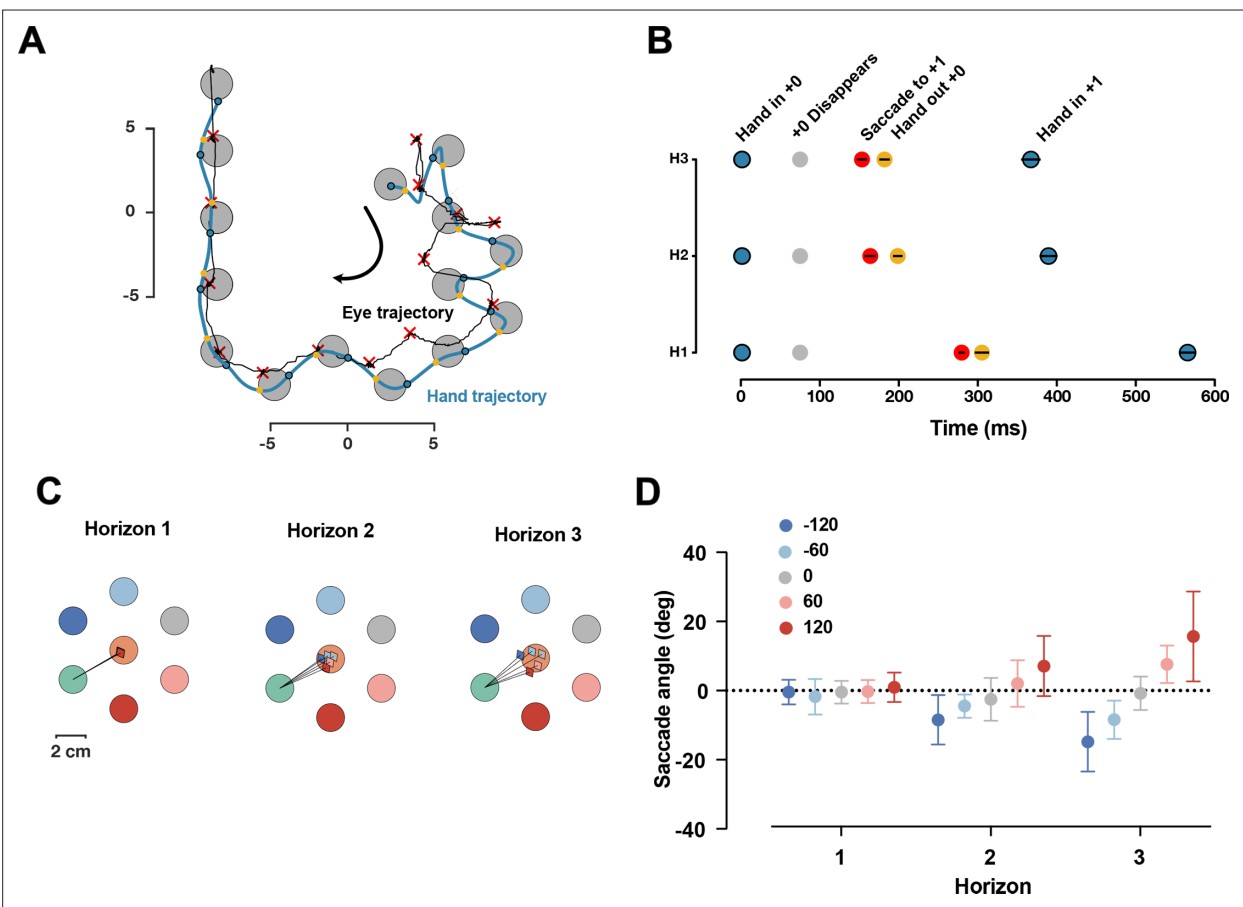

**Figure 7.** Saccade position is shifted toward the future target. (**A**) One sample trial from a representative participant. The blue and black trace shows the hand and eye position, respectively. The blue dots are when hand first entered the target, yellow dots show when hand exited the target, the red × shows the fixation location. (**B**) Timing of capturing the current target (gray), saccade to the next target (red), hand out of the captured target (yellow), and hand in the next target (blue), averaged across participants, black lines show 95% confidence interval. (**C**) Saccade angles of a representative participants for each possible +2 target and three horizons. (**D**) Average saccade angle for all participants (n = 19) and each horizon. Error bars show 95% confidence interval.

curvature effect was smaller ($F_{(2,20)}$ = 16.71, p=6.00e-04), suggesting less interaction between planning processes when the movements are biomechanically separated.

Together, these results show that the biomechanically advantageous co-articulation between segments of the sequence occur when the segments are planned together.

### Fixation location is modified by the availability of future targets

Given that participants used visual information from two targets ahead, we were curious whether the availability of future targets influences participants' eye-movement strategy, or whether they acquired this information parafoveally.

We collected data from a separate group of 19 participants in the same task, only focusing on Dwell time 75 ms, and H1, H2, and H3 conditions. We found that in ~95% of all reaches in all trials, participants made only one saccade per reach, indicating they primarily focused on the immediate next target with information about subsequent targets likely being processed through parafoveal vision. We assessed the timing of the saccades by measuring the saccade time relative to hand position (*Figure 7A and B*). The saccade to the next target occurred after the current target disappeared, and right before the hand exited the disappeared target. The relative timing between the saccade and the hand exiting the target was consistent across different horizons ($F_{(2,36)}$ = 1.92, p=0.16).

Next, we characterized the fixation location. If the visual information of the next target is received through the parafoveal vision, then it would be beneficial to shift the fixation location toward the future target. As shown in one representative participant (*Figure 7C*), we found that saccade angle is shifted toward the position of the future target in the H2 and H3 conditions. At the group level, the saccade angle was significantly different for different horizons ($F_{(8,128)}$ = 12.60, p=3.12e-13). In Horizon 1, as expected, the average saccade angle for different targets was identical since the future target was not shown. For both Horizon 2 and Horizon 3, the angle of the saccade was systematically shifted toward the position of the upcoming target.

Together, these results suggest that information about future targets is received parafoveally, and the fixation location is systematically shifted to facilitate this process.

## Discussion

### Planning horizon in sequential reaching versus finger presses

In a previous finger sequence study, using a similar horizon manipulation to the one used here, we found that participants executed sequences faster when they had information about multiple future finger presses (*Ariani et al., 2021*). This benefit increased up to horizon of three future finger presses (H3) and then plateaued. Consistent with these findings, we observed a large reduction in movement time when participants were provided with one future reach target (*Figure 2*, H1 to H2). However, except for the shortest (75 ms) dwell time, the availability of a second future target (H2 to H3) did not further reduce movement time (*Figure 2*). In these longer (200 and 400 ms) dwell times, we did not observe faster performance for more knowledge of future target positions, likely because the participants had sufficient time to complete planning during the dwell period. Another possible reason for the more pronounced effect of horizon on movement speed in finger presses may be attributed to the nature of the effectors. Specifically, in reaching movements, the arm cannot initiate the next reach before completing the previous one. In contrast, with finger movements, future finger flexions can commence in advance, potentially resulting in faster execution of the sequence (*Popp et al., 2022*). It is also possible that this difference arises because the transformation of the visual cue to motor plans is faster for the direct spatial mapping used here than for the more abstract number-to-finger mapping used in our previous study (*Diedrichsen et al., 2001*; *Goodman and Kelso, 1980*). Even though the planning of multiple future movements, as measured by IRI, could only be seen in shortest dwell times, our experiments with target displacements provide clear evidence that participants had planned two movements ahead (*Figure 3*). Overall, these observations suggest that the availability of the second reach target can be more significant when faster execution of the task demands faster transformation of visual cue to muscle commands, or when the cue-to-action mapping is more demanding.

## Interactions among future movement plans

If participants plan multiple future movements at the same time, the next question is whether these preparatory processes run independently or if they interact with each other. We investigated these possibilities by jumping the target that participants were about to reach toward. The participants corrected the reach only after initially reaching toward the pre-jump position of the target (*Figure 4B*). This behavior is similar to that shown in the work by *Ames et al., 2019*, where target displacement during execution led to an initial commitment to the pre-jump position of the target followed by a smooth corrective reach toward the new target position. Neurally, the authors showed that resource distribution in M1 and PMd is accomplished by re-planning the corrective reach in a subspace orthogonal to the one controlling the ongoing movement (*Ames et al., 2019*). Here, we asked whether the re-planning process depended on any other planned future movement (*Figure 4*). Interestingly, the corrections were slower when more future targets were known to the participants (*Figure 4B*), indicating some interaction between the two future planned reaches. This interaction could come in multiple forms. One possibility is that the neural resources dedicated to re-planning have to be split between preparation of future targets, slowing the re-planning of the next movement (*Kornysheva et al., 2019*). Alternatively, the two future movements may be prepared as a chunk (*Ramkumar et al., 2016*), and changing the entire chunk may take longer time than changing a single movement. The latter possibility seems unlikely since the results from jumping the +1 or +2 target within the same horizon of future target can be corrected separately (*Figure 5*). Either way, by probing the planned state with target perturbations we clearly demonstrate an obligatory interaction between multiple future movement plans.

We also provide evidence that the interactions between future movement plans can optimize kinematics of single reaches for the next reach in the sequence. When the planning processes of two future reaches overlapped sufficiently, we found changes in the curvature of the current reach that anticipated the direction of the next reach target. The curvature was opposite to the direction of the next target, making this co-articulation advantageous from a biomechanical point of view (*Figure 6D*). The observed curvature interaction can again be either due to fully chunked planning of two elements, or alternatively, due to separate, yet interactive, planning of the two reaches. The former possibility seems less likely since the interaction was observed even when movement segments were fully separated by a long dwell time (*Figure 6D*, Dwell 400).

## Implications for the neural control of online planning

What implications do our results have for the neural processes underlying the online planning of multiple future actions? Previous neurophysiological investigations showed that individual neurons can be involved in both the planning and execution of phases of a movement (*Churchland and Shenoy, 2007*; *Crammond and Kalaska, 2000*; *Elsayed et al., 2016*; *Kaufman et al., 2014*; *Pruszynski et al., 2014*). Nonetheless, when two movements are concatenated, the planning of the second movement can be proceeded in parallel with the control of the first movement. This lack of interference can be explained by the fact that planning and execution proceed in orthogonal neural subspaces (*Zimnik and Churchland, 2021*).

The phenomena demonstrated in this study raise the question of how the planning processes for multiple future movements are realized in the brain. One hypothesis is that the two future movements are also planned in orthogonal neural subspaces without any interactions during the planning phase. Under this hypothesis, the co-articulation we report would arise from an interaction between the execution dynamics associated with the current movement and the planning dynamics of the second planned movement. An alternative hypothesis is that the preparation processes of the next two movements directly interact with each other, and possibly are even encoded in partly together (*Fusi et al., 2016*; *Rigotti et al., 2013*). Our results are suggestive of the latter scheme since we observed no co-articulation when the next target was only available during execution of the current reach (*Figure 6D*, H2). Nevertheless, careful electrophysiology experiments are necessary to investigate the exact mechanism by which planning processes interact. The current paradigm provides a useful framework to do so.

## Eye movement coordination during sequence production

In our sequence task, participants switched their gaze location only once per reach, suggesting that information about the location of the next target is perceived parafoveally (*Figure 7A*). This

observation aligns with previous studies (*Clavagnier et al., 2007*; *González-Alvarez et al., 2007*; *Sivak and MacKenzie, 1990*) that found participants keep their visual attention on the current sequence item and can perceive the location of spatial targets even when foveal vision is occluded. However, when comparing gaze locations for conditions Horizon>1, we observed that participants systematically biased their gaze location based on the sequence context. The gaze position shifted toward the next target, potentially allowing for more accurate location estimation (*Figure 7C and D*). Notably, changes in gaze location were observed even in Horizon 2, despite no changes in the curvature of hand movements in this horizon (*Figure 6B*). This suggests that information about the next target may first be available in the circuitry that controls eye movements and later in the cortical areas that control voluntary upper limb movements. Further control studies are required to investigate this hypothesis.

## Methods
### Participants
Eleven participants (4 female) with an average age of 23.3 years (4.4 SD years) completed five experimental visits for this study (~10 hr data collection per participant); this data was used for IRI (*Figure 2*) and curvature analysis (*Figure 6*). Ten of these participants returned for two experimental visits where they were tested on target jump experiments. They were all right-handed with average handedness of 78 (24 SD), measured by the Edinburgh Handedness Inventory. For eye position analysis, we recruited a different group of 20 right-handed participants (2 female) with average age of 21.2 years (2.0 SD years), one participant was excluded from the analysis due to low quality of data (see Eye-tracker analysis below).

All participants reported no prior history of musculoskeletal, neurological, or psychiatric disorders. All the participants provided informed consent in the first session, and they were remunerated CA\$ 15 per hour in the seventh and last session of the study. All the procedures were approved by the Health Sciences Research Ethics Board at the University of Western Ontario (Project ID 115088).

### Apparatus
Participants performed all experimental trials in an exoskeleton robot (Kinarm, Kingston, ON, Canada). The participants were seated on a height-adjustable chair while their right arm rested comfortably on the robot arm, which supported the elbow and shoulder weight against gravitational force and allowed them to freely move their hand in the horizontal plane. Arm kinematics were recorded at 1000 Hz. All the reaching targets were presented by a horizontally placed monitor onto a mirror which occluded the vision of the participant's arm (*Figure 1A*). Participants' eye movement was recorded using an SR Research Eyelink 1000 at 1000 Hz. The eye tracker also recorded the participants' head movement by recording the position a bullseye target attached to the participants' front. The eye position was subtracted from the bullseye position to correct for small head movements during the task.

### General procedures
In each experimental trial, participants performed sequences of 14 reaches. The sequences were generated from a hexagonal grid of equidistant circular targets (see *Figure 1A and B*). The radii of the targets were 1 cm, and the center of neighboring targets was 4 cm apart. The participants' arm was occluded – they only saw a circle with radius of 0.5 cm aligned with the tip of their index finger as their hand feedback. The sequences always started from a fixed home target in the center of the working space. We generated sequences according to two rules. First, the next target in the sequence should be a neighbor of the previous target. This ensured that all the reaches were 4 cm apart. Second that there were no loops smaller than five reaches. This ensured that, when multiple future targets were presented, they did not overlap. The participants were instructed to move their right hand in the home target to start a trial. Once the hand was in the home target, either one, two, three, four, or five future targets of the sequence appeared on the screen (depending on the horizon condition); brightness indicated the order of targets, with the brightest target being the immediate next target. The participants were instructed to stay in the home target for 300 ms, after which they received a go cue by the disappearance of the home target. The participants were instructed to always move their hand to the brightest target and stay in the target until it was 'captured'. Once one target was captured,

the captured target disappeared, the brightness of the targets was updated, making the next target the brightest, and a new target appeared at the end of the horizon. This process was repeated until all 14 targets were captured. If the participant failed to stay in the target for the dwell time or the initial wait time in the home target, the trial was interrupted with an error message and rejected. Interrupted trials were repeated later in the session.

Our experiments manipulated two parameters: how much time participants had to stay in each target to capture it (dwell time), and how many future targets were on the screen (horizon). The dwell times could be 75, 200, or 400 ms. In the horizon conditions (H1 to H5), 1–5 future targets were visible. In the case of H1, the task is reduced to a sequential reaction time task, and with longer horizons, participants could potentially plan multiple future movements ahead of time.

The entire experiment had seven sessions. The first five sessions were designed to get a time and curvature analysis in all dwell and horizon conditions. The last two session added the jump experiment.

## Time and curvature analysis

The first five sessions measured performance in 15 conditions (3 dwell times × 5 horizons). Each session consisted of three blocks of 120 trials for each dwell time, and the horizon was randomized across trials totaling 360 trials per session. The order of dwell time blocks was randomized across five sessions for each participant. Each session of data collection was 1 hr and 15 min on average. As the first step, for each trial, we broke down the full sequence of reaches to their constituting single reaches by segmenting the full sequence trajectory whenever a target was captured. This led to a set of 14 individual reaches starting from each target and ending in the next. For all the analysis we were interested in simultaneous planning and execution processes, therefore we excluded all targets that were visible in the beginning and could therefore be pre-planned. We also excluded one to five targets at the end of each sequence since there was no need to plan future targets anymore. The number of excluded reaches changed with the horizon. For instance, in the H1 condition, we excluded the first and the last reach in the sequence.

Given our hexagonal grid, for each reach, there could be a maximum of five potential next target positions. However, near the boundaries of the workspace, the number of potential next targets decreases so the participants could potentially predict the overall position of the upcoming target and plan for it ahead of time (*Glaser et al., 2018*). To ensure that this possibility did not affect our results, we only considered reaches with five potential future choices. This excluded the reaches toward and parallel to the boundaries of the workspace.

For the analysis of movement time, we computed IRI, defined as the time the hand entered a one target until it entered the next one. We subsequently averaged IRI values across all the reaches of a trial, all trials, and all sessions of each participant. The IRI contains both the time that the hand passed through the target and the time that the hand was moving between the two targets.

For the curvature analysis, we assessed the effect of the position of the +2 target on the curvature of the reach toward the +1 target. We started by aligning all the reaches: First, we translated the position of the +0 target (where the hand is sitting), the −1 target, the +1 target, and the reach trajectory so that the position of the +0 target is set to the center of the 2D coordinate system (0,0) cm. This ensures that all the reaches start from the same position. Next, we rotated the targets and trajectory around the +0 target so the position of the +1 target rests at (4,0) cm coordinates. This ensures that all the reaches have the same directions. With these transformations, the angle of the line connecting the +1 target to the +2 target, relative to the horizon line, connecting the +0 target to the +1 target, can be either −120,−60, 0, 60, or 120 degrees. The same is true for the angle of the line connecting the −1 target to the +0 target.

Next, we quantified the curvature of +1 reach. We used all the translated and rotated reaches of all the participants. To make the length of the reaches equal, for each reach, we took 100 equally distant spatial samples along the horizontal line connecting the center of the start target to the center of the end target of the reach. Then, we performed a PCA on the matrix containing the y coordinate values of each reach. The size of this matrix was (# reaches × 100). The first and the second PCs were arc-shaped and S-shaped 'eigen reaches' each accounting for 72% and 17% of the total variance. We then projected each reach onto the first eigen reach and used the resultant scalar value as a measure of curvature. The absolute value of this scalar shows the amount of the curvature, and the sign indicates the direction of the curve. We used all the reaches of all the participants to calculate the PCs, and then

for each participant and condition, the curvature value was calculated separately. This ensured that the comparison between the conditions and averaging across participants are meaningful.

Finally, we were interested in the effect of the +2 target angle on the curvature of the reach to the +1 target. However, the curvature of a reach in a sequence also depends on the previous reach, therefore, to account for this effect, we fitted a linear model that predicted the signed curvature value of each reach based on the position of the previous target (angle of −1 target), next target (angle of +2 target). The angles were one-hot coded, resulting in one regressor for each angle; therefore, the beta values represent the effect of each input angle onto the curvature effect. This process was performed for each of the dwell time and horizon conditions separately. Finally, as summary statistics for the effect of all the one-hot coded values the outgoing target (+2 target effect), we fitted a line to beta values for each of the five angles. We used the slope of this line as a summary of the overall effect. Zero slopes indicated no curvature effect, the value and sign of the slope show the strength and direction of the effect, with a negative slope showing curvature toward the opposite direction of the next target, and vice versa.

## Jump experiment

For the last two data collection sessions, we focused on the 75 ms dwell condition and two of the horizons (H2, H3). All other parameters including the grid of targets, length of the sequence, size of the targets, etc. were identical to the first five sessions. In these experiments, only one jump of a target could happen in each trial. The jump happened randomly between the 4th and 10th reach of the sequence. We interleaved many no-jump trials in these sessions to avoid anticipation or adaptations for the jumps. The order of these two last sessions was randomized across participants.

The +1 jump experiment consisted of 400 trials, 200 target jumps in H2 and H3, interleaved with 200 no-jump trials with randomized across horizons. In the case of a jump trial, we displaced the next target (+1 target) exactly when the current target (+0 target) was captured. Before the jump, a pre-jump +1 target was shown on the screen, and then, at the moment of the jump, that is when the 75 ms dwell time was satisfied and the current target (+0 target) was captured, we removed the pre-jump +1 target and a new +1 target appeared on the screen. Both the new and pre-jump +1 target were selected in a way that was compatible with the current position of the +2 or +3 targets on the screen, in other words, the jump was compatible with the rules of generating sequence in the task. This jump happened both in the context of Horizon 2, with two future targets, and in Horizon 3, with three future targets presented on the screen.

The +2 jump experiment was performed only in Horizon 3 (H3) condition. There was a total of 300 trials. Two-third of them were no jump; in the remaining one-third, exactly at the movement that the current target (+0 target) was captured, the second future target (+2 target) jumped to a new position, and the position of the next target (+1 target) remained unchanged. Before the jump, we showed a temporary +2 target (pre-jump +2 target) on the screen, and the jump happened with the disappearance of the pre-jump +2 target and the appearance of a new target as the new +2 target.

## Eye-tracker analysis

For pre-processing of eye-tracker data, we first removed trials if the eye position was not recorded for a consecutive 600 ms. This missing data could be due to blinks or the eye tracker momentarily losing the eye position. We removed one participant due to low number of good trials. Then for the remaining trials we first partitioned each trial to 14 segments based on the time point that participants' hand entered a new target. Then, within each segment we applied a fixed threshold on the derivative of eye position to detect the time point when the saccade occurred. Next, we used the average eye position before and after saccade for analysis of the saccade position. In 95% of the trial only one saccade happened in each reach. The eye position before and after saccade was centered on a circle with radius of 2 cm around the start and end target, respectively. These pre-processing steps were performed blind to the trial type. To analyze the changes in saccade position based on horizon, like curvature analysis, for each participant we first translated and rotated all the reaches and eye positions so that the start target (+0 target) and end target (+1 target) of the reaches are at (0,0) and (4,0) coordinate, respectively. Then for each horizon, we averaged the post-saccade eye position for each possible +2 target position.

## Statistical analysis

We employed a within-subject design. All the analyses were performed in RStudio 22.07.1. For analysis of IRI and curvature effect, we used two-way repeated measures ANOVA. Factors were dwell time (three levels), horizon (five levels: H1 to H5), and (dwell time × horizon) interaction. For comparison between different levels of each significant factor, we adjusted p-values for multiple comparisons using Holm method. For the jump +1 target experiment, we used a repeated measures two-way ANOVA with jump (two levels) and horizon (two levels) as factors. Correction for multiple comparisons was similar to the IRI analysis. The details of statistical analysis including the degrees of freedom, the test statistic, and the p-value are provided in the text.

## Acknowledgements

This work was supported by a CIHR Project Grant to JD and JAP (PJT-175010). JAP received a salary award from the Canada Research Chairs Program.

## Additional information

### Competing interests

J Andrew Pruszynski: Reviewing editor, eLife. The other authors declare that no competing interests exist.

### Funding

| Funder | Grant reference number | Author |
| --- | --- | --- |
| Canadian Institutes of Health Research | PJT-175010 | Jörn Diedrichsen<br>J Andrew Pruszynski |
| Canada First Research Excellence Fund | | Jörn Diedrichsen<br>J Andrew Pruszynski |
| Canada Research Chairs | | J Andrew Pruszynski |
| Canada Foundation for Innovation | | J Andrew Pruszynski |

The funders had no role in study design, data collection and interpretation, or the decision to submit the work for publication.

### Author contributions

Mehrdad Kashefi, Conceptualization, Data curation, Formal analysis, Visualization, Methodology, Writing – original draft, Writing – review and editing; Sasha Reschechtko, Giacomo Ariani, Conceptualization, Writing – review and editing; Mahdiyar Shahbazi, Writing – review and editing; Alice Tan, Data curation, Writing – review and editing; Jörn Diedrichsen, Conceptualization, Data curation, Supervision, Methodology, Writing – original draft, Writing – review and editing; J Andrew Pruszynski, Conceptualization, Resources, Supervision, Methodology, Writing – original draft, Writing – review and editing

### Author ORCIDs

Mehrdad Kashefi ● https://orcid.org/0000-0001-5981-5923
Sasha Reschechtko ● https://orcid.org/0000-0003-1025-4533
Giacomo Ariani ● https://orcid.org/0000-0001-9074-1272
J Andrew Pruszynski ● https://orcid.org/0000-0003-0786-0081

### Ethics

All the participants provided informed consent in the first session, and they were remunerated CA$ 15 per hour in the seventh and last session of the study. All the procedures were approved by the Health Sciences Research Ethics Board at the University of Western Ontario (Project ID 115088).

Reviewer #1 (Public Review): https://doi.org/10.7554/eLife.94485.3.sa1
Reviewer #2 (Public Review): https://doi.org/10.7554/eLife.94485.3.sa2
Author response https://doi.org/10.7554/eLife.94485.3.sa3

## Additional files

### Supplementary files
• MDAR checklist

### Data availability

All the raw data generated as part of this study are publicly available. The data has been uploaded to Dryad: https://doi.org/10.5061/dryad.7pvmcvf30.

The following dataset was generated:

| Author(s) | Year | Dataset title | Dataset URL | Database and Identifier |
| --- | --- | --- | --- | --- |
| Kashefi M, Reschechtko S, Ariani G, Shahbazi M, Tan A, Diedrichsen J, Pruszynski JA | 2024 | Data from: Future movement plans interact in sequential arm movements | https://doi.org/10.5061/dryad.7pvmcvf30 | Dryad Digital Repository, 10.5061/dryad.7pvmcvf30 |

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
