## [Editor Report · eLife assessment]

This study presents an **important** set of results illuminating how movement sequences are planned. Using several different behavioural manipulations and analysis methods, the authors present **compelling** evidence that multiple future movements are planned simultaneously with execution, and that these future movement plans influence each other. The work will be of great interest to those studying motor control.

---

## [Referee Report · Reviewer #1 (Public Review)]

Mehrdad Kashefi et al. investigated the availability of planning future reaches while simultaneously controlling the execution of the current reach. Through a series of experiments employing a novel sequential arm reaching paradigm they developed, the authors made several findings: (1) participants demonstrate the capability to plan future reaches in advance, thereby accelerating the execution of the reaching sequence, (2) planning processes for future movements are not independent one another, however, it's not a single chunk neither, (3) Interaction among these planning processes optimizes the current movement for the movement that comes after for it.

The question of this paper is very interesting, and the conclusions of this paper are well supported by data.

---

## [Referee Report · Reviewer #2 (Public Review)]

In this work, Kashefi et al. investigate the planning of sequential reaching movements and how the additional information about future reaches affects planning and execution. This study, carried out with human subjects, extends a body of research in sequential movements to ask important questions: How many future reaches can you plan in advance? And how do those future plans interact with each other?

The authors designed several experiments to address these questions, finding that information about future targets makes reaches more efficient in both timing and path curvature. Further, with some clever target jump manipulations, the authors show that plans for a distant future reach can influence plans for a near future reach, suggesting that the planning for multiple future reaches is not independent. Lastly, the authors show that information about future targets is acquired parafoveally--that is, subjects tend to fixate mainly on the target they are about to reach to, acquiring future target information by paying attention to targets outside the fixation point.

The study opens up exciting questions about how this kind of multi-target planning is implemented in the brain. As the authors note in the manuscript, previous work in monkeys showed that preparatory neural activity for a future reaching movement can occur simultaneously with a current reaching movement, but that study was limited to the monkey only knowing about two future targets. It would be quite interesting to see how neural activity partitions preparatory activity for a third future target, given that this study shows that the third target's planning may interact with the second target's planning.

[Editors' note: The authors fully addressed the reviewers' comments on the original manuscript.]

---

## [Author Response]

The following is the authors’ response to the original reviews.

**Public Reviews:**

**Reviewer #1:**
Mehrdad Kashefi et al. investigated the availability of planning future reaches while simultaneously controlling the execution of the current reach. Through a series of experiments employing a novel sequential arm reaching paradigm they developed, the authors made several findings: (1) participants demonstrate the capability to plan future reaches in advance, thereby accelerating the execution of the reaching sequence, (2) planning processes for future movements are not independent one another, however, it's not a single chunk neither, (3) Interaction among these planning processes optimizes the current movement for the movement that comes after for it.The question of this paper is very interesting, and the conclusions of this paper are well supported by data. However, certain aspects require further clarification and expansion.

We thank reviewer one for their evaluation of the work.

(1) The question of this study is whether future reach plans are available during an ongoing reach. In the abstract, the authors summarized that "participants plan at least two future reaches simultaneously with an ongoing reach and that the planning processes of the two future reaches are not independent of one another" and showed the evidence in the next sentences. However the evidence is about the relationship about ongoing reach and future plans but not about in between future plans (Line 52-55). But the last sentence (Line 55-58) mentioned about interactions between future plans only. There are some discrepancies between sentences. Could you make the abstract clear by mentioning interference between (1) ongoing movement and future plans and (2) in between future plans?

We thank Reviewer for their comment. We have separated the longer sentence in the original abstract into two shorter ones. This should clarify that the two pieces of evidence pertain to the interaction of planning processes.

(2) I understood the ongoing reach and future reaches are not independent from the results of first experiment (Figure 2). A target for the current reach is shown at Horizon 1, on the other hand, in Horizon 2, a current and a future target are shown on the screen. Inter-reach-interval was significantly reduced from H1 to H2 (Figure 2). The authors insist that "these results suggest that participants can plan two targets (I guess +1 and +2) ahead of the current reach (I guess +0)". But I think these results suggest that participants can plan a target (+1) ahead of the current reach (+0) because participants could see the current (+0) and a future target (+1) in H2. Could the authors please clarify this point?

We thank Reviewer for raising this point. Our conclusion that “participants can plan two targets ahead of the current reach” is supported by the reduction in Inter-Response Interval (IRI) observed when comparing H2 to H3 in the 75 ms Dwell time condition. Specifically, on average, participants were 16 ms faster when they could see two future targets on the screen (H3) than when they could see only one (H2). To clarify this in the paper, we have revised the wording in line 124 to explicitly state that the conclusion pertains to the 75 ms Dwell time condition. Additionally, we emphasize that the strongest evidence for planning two future targets comes from the experiment shown in Figure 3.

(3) Movement correction for jump of the +1 target takes longer time in H3 compared to H2 (Figure 4). Does this perturbation have any effect on reaching for +2 target? If the +1 jump doesn't affect reaching for +2 target, combined with the result that jump of the +2 target didn't affect the movement time of +1 target (Figure 3C), perturbation (target jump) only affects the movement directly perturbed. Is this implementation correct? If so, does these results support to decline future reaches are planned as motor chunk? I would like to know the author's thoughts about this.

In the experiment presented in Figure 4, once we jumped the +1 target, the reach to that target was changed and participants replaned a corrective movement to the new location of the +1 target. This usually was followed by a longer-than-usual pause at the new location of +1 target for resuming the sequence and finishing the trial. Consequently, in these jump trials, it was impossible to compare the +2 reach to no-jump trials, as the normal sequence of movement was disrupted, and the reach to the +2 target originated from a different starting location. Nevertheless, we addressed the possibility that the two future reaches were planned as a chunk by the analysis shown in figure 5: There we showed that a displacement of the +2 target did not influence the reach to the +1 target, indicating that the movement plans could be updated independently.

(4) Any discussion about Saccade position (Figure 7)?

We thank reviewer 1 for this important comment. The following discussion section is added for the gaze position results.

In our sequence task, participants switched their gaze location only once per reach, suggesting that information about the location of the next target is perceived parafoveally (Figure 7A). This observation aligns with previous studies (Clavagnier et al., 2007; González-Alvarez et al., 2007; Sivak and MacKenzie, 1990) that found participants keep their visual attention on the current sequence item and can perceive the location of spatial targets even when foveal vision is occluded. However, when comparing gaze locations for conditions Horizon >1, we observed that participants systematically biased their gaze location based on the sequence context. The gaze position shifted toward the next target, potentially allowing for more accurate location estimation (Figures 7C-D). Notably, changes in gaze location were observed even in Horizon 2, despite no changes in the curvature of hand movements in this horizon (Figure 6B). This suggests that information about the next target may first be available in the circuitry that controls eye movements and later in the cortical areas that control voluntary upper limb movements. Further control studies are required to investigate this hypothesis.

**Reviewer #2:**
Summary:In this work, Kashefi et al. investigate the planning of sequential reaching movements and how the additional information about future reaches affects planning and execution. This study, carried out with human subjects, extends a body of research in sequential movements to ask important questions: How many future reaches can you plan in advance? And how do those future plans interact with each other?The authors designed several experiments to address these questions, finding that information about future targets makes reaches more efficient in both timing and path curvature. Further, with some clever target jump manipulations, the authors show that plans for a distant future reach can influence plans for a near future reach, suggesting that the planning for multiple future reaches is not independent. Lastly, the authors show that information about future targets is acquired parafoveally--that is, subjects tend to fixate mainly on the target they are about to reach to, acquiring future target information by paying attention to targets outside the fixation point.The study opens up exciting questions about how this kind of multi-target planning is implemented in the brain. As the authors note in the manuscript, previous work in monkeys showed that preparatory neural activity for a future reaching movement can occur simultaneously with a current reaching movement, but that study was limited to the monkey only knowing about two future targets. It would be quite interesting to see how neural activity partitions preparatory activity for a third future target, given that this study shows that the third target's planning may interact with the second target's planning.Strengths:A major strength of this study is that the experiments and analyses are designed to answer complementary questions, which together form a relatively complete picture of how subjects act on future target information. This complete description of a complex behavior will be a boon to future work in understanding the neural control of sequential, compound movements.

We thank the reviewer for their thorough reading of our work.

Weaknesses:I found no real glaring weaknesses with the paper, though I do wish that there had been some more discussion of what happens to planning with longer dwell times in target. In the later parts of the manuscript, the authors mention that the co-articulation result (where reaches are curved to make future target acquisition more efficient) was less evident for longer dwell times, likely because for longer dwell times, the subject needs to fully stop in target before moving to the next one. This result made me wonder if the future plan interaction effect (tested with the target jumps) would have been affected by dwell time. As far as I can tell, the target jump portion only dealt with the shorter dwell times, but if the authors had longer dwell time data for these experiments, I would appreciate seeing the results and interpretations.

We thank the reviewer for raising this point. In our time (Figure 2) and curvature analysis (Figure 6), we collected data with five levels of the horizon and three levels of dwell time to explore the space of parameters and to see if there is any interaction between dwell time and the horizon of planning the future targets. Apriori, we expected that the full stop in each target imposed by the 400 ms dwell time would be long enough to remove any effect of future targets on how the current move is executed. In line with our initial hypothesis, the systematic curvature of reaches based on the future target was smaller in longer dwell times (Figure 6E). Nevertheless, we observed a significant curvature even in 400 ms dwell time. Based on this observation, we expect running the jump experiments (Figures 4 and 5) in longer dwell times will lead to the same pattern of results but with a smaller effect size since longer dwells break the interdependence of sequence elements (Kalidindi & Crevecoeur, 2023). In the end, for the jump experiments, we limited our experimental conditions to the fastest dwell time (75 ms dwell) since we were conceptually interested in situations where movements in the sequence are maximally dependent on each other.

Beyond this , the authors also mentioned in the results and discussion the idea of "neural resources" being assigned to replan movements, but it's not clear to me what this might actually mean concretely. I wonder if the authors have a toy model in mind for what this kind of resource reassignment could mean. I realize it would likely be quite speculative, but I would greatly appreciate a description or some sort of intuition if possible.

Our use of the term "neural resources" is inspired by classic psychology literature on how cognitive resources such as attention and working memory are divided between multiple sequence components. Early studies on working memory suggest that human participants can retain and manipulate a fixed number of abstract items in working memory (Miller, 1956). However, more recent literature postulates that a specific number of items does not limit working memory, rather, it is limited by a finite attentional resource that is softly allocated to task items.

Here we borrowed the same notion of soft distribution of resources for the preparation of multiple sequence items. A large portion of our observation in this paper and also previous work on sequence production can be explained by a simple model that assumes one central planning resource that is “softly” divided between sequence elements when participants see future items of the sequence (Author Response Image 1). The first sequence element receives the majority of the resources and is planned the most. The rest of the sequence receives the remaining planning resources in an exponentially decaying manner for preparation of the movement during the execution of the ongoing movement. Once the ongoing movement is over, the resource is then transferred to the next sequence item and this process is repeated until the sequence is over. Assignment of planning resources to future items explains why participants are faster when seeing future items (Figure 2). But this comes with a cost – if the ongoing movement is perturbed, the replanning process is delayed since some of the resources are occupied by future planning (Figure 4). This naturally leads to the question of how this resource allocation is implemented in neural tissue. To address this, we are conducting the same sequence task with the horizon in non-human primates (NHPs), and the investigation of these neural implementation questions will be the focus of future studies.

**Author response image 1. sa3fig1:** Basic diagram showing a soft distribution of a limited planning resource.

**Recommendations for the author:**

**Reviewer #1**

We thank reviewer one for these comments regarding the clarity and consistency of figures and terminology.

(1) Figure 3. Are "+1 Move" in Fig. 3B and "+ 1 Movement" in Fig. 3C as same as "E + 1" in Fig. 3A? Also does "Dwell" in Fig. 3B mean same as "+1 Dwell" in Fig. 3C? Consistent terminology would help readers to understand the figure.

“+1 Move” in Figure 3B is the same as +1 movement in Figure 3C. “Dwell” in Figure 3B is the same as +1 Dwell in Figure 3C. We changed the figure for more consistency.

(2) Figure 3. A type in the second last line in the legend, "pre-jump target for no-jump and jump and condition". The second "and" isn't necessary.

The typo is corrected. Thank you.

(3) Figure 4C. Is "Movement time" equivalent with "E + 1"?

“Movement time” is equivalent to E+1 only in no-jump conditions. When the jump occurs,

Movement time contains all the

(4) Figure 6B. Is the gray circle in between the graph and target positions there by mistake?

We fixed this typo. Thank you.

(5) Figure 6E. It's hard to distinguish H2-H5 from the color differences.

We changed the H5 to full white with a black stroke to improve the contrast. Thank you.

(6) Figure 7A. Blue dots are almost invisible.

We added a black stroke to blue circles for more visibility. Thank you.

**Reviewer #2**
I found this manuscript to be engaging and well written--many of the questions I had while reading were answered promptly in the next section. As such, my comments are mostly minor and primarily geared towards improving clarity in the manuscript.(1) One major recurring confusion I had while reading the manuscript was how to think about H1, H2, and H3. It was clearly explained in the text, and the explanations of the results were generally clear once I read through it all, but I found it strangely confusing at times when trying to interpret the figures for myself (e.g., in H2, 2 targets are on screen, but the second target can only be planned during the reach toward the first target). This confusion may just be me reading the manuscript over two days, but I wonder if it could be made clearer with some semantic iconography associated with each horizon added to the later figures alongside the H labels. As one option, perhaps the planning timeline part of Fig 1D could be simplified and shrunk down to make an icon for each horizon that clearly shows when planning overlaps for each horizon.

(Please see the response to point #2 below)

(2) Regarding Fig 1D: I like this figure, but it's unclear to me how the exact preparation and execution times are determined. Is this more of a general schematic of overlaps, or is there specific information about timing in here?

We thank reviewer 2 for their important feedback. The role of Figure 1D was to summarize the timing of the experiments for different horizons. That is, to clarify the relative timing of the targets appearing on the screen (shown with a small circle above the horizontal line) and targets being captured by participants (the ticks and their associated number on the line). Execution is shown as the time interval that the hand is moving between the targets and planning is the potential planning time for participants from the target appearing on the screen until initiation of the reach to that target. We added the relevant parts of Figure 1D to the subplots for each subsequent experiment, to summarize the timing of other experiments and their analyses. For the experiments with target jump, a small vertical arrow shows the time of the target jump relative to other events.

However, this figure will be less useful, if the connection between the timing dots and ticks is not communicated. We agree that in the original manuscript, this important figure was only briefly explained in the caption of Figure 1. We expanded the explanation in the caption of Figure 1 and referenced the dots and ticks in the main text.

(3) Fig 6B - for some reason I got confused here: I thought the central target in this figure was the start target, and it took me embarrassingly long to figure out that the green target was the start target. This is likely because I'm used to seeing center-out behavioral figures. Incidentally, I wasn't confused by 7c (in fact, seeing 7c is what made me understand 6b), so maybe the solution is to clearly mark a directionality to the reach trajectories, or to point an arrow at the green target like in previous figures. Also, the bottom left gray target in the figure blends into the graph on the left--I didn't notice it until rereading. Because there's white space between that target and the green one, it might be good to introduce some white space to separate the graph from the targets more. The target arrangement makes more sense in panel C, but by the time I got there, I had already been a bit confused.

Thanks for raising this point. As shown in Figure 6C, we used the reach to the +1 target for the curvature analysis. The confusion about Figure 6B is probably due to continuing the reach trajectories after the +1 target. That also explains why Figure 7C seemed more straightforward. To solve this issue we modified Figure 6B such that the reaches are shown with full opacity right until the +1 target and then shown with more transparency. We believe this change focuses the reader's attention to the reach initiated from the +0 target to the +1 target.

As for the gray target in Figure 6B, we originally had the gray target as it is a potential start location for the reach to the +0 target, and for having similar visuals between the plots. The gray target is now removed from Figure 6B.

(4) Line 253 - I'm not sure I understand the advantage over simple averaging that the authors mention here--would be nice to get a bit more intuition.

Thanks for raising this point. We used a two-factor model in our analysis, with each factor representing the angle of the last and next target, respectively. Both factors had five levels: -120, -60, 0, 60, and 120 degrees relative to the +1 reach. In a balanced two-factor design, where each combination of factor levels has an equal number of trials, using a linear model and simple averaging would yield equivalent results. However, when the number of trials for the combinations of the two factors is unbalanced, simple averaging can lead to misleading differences in the levels of the second factor. Additionally, the linear model allows us to investigate potential interactions between the two factors, which is not possible with simple averaging.

(5) Fig 7a - I would have liked to see the traces labeled in figure (i.e. hand trajectory vs. eye trajectory)

Hand and eye trajectories are now labeled in the figure.

(6) Fig 7c - very minor, but the hexagon of targets is rotated 30 degrees from all previous hexagons shown (also, this hex grid target arrangement can't lead to the trajectory shown in 7a, so it can't be that this was a different experimental grid). I'm guessing this was a simple oversight.

We used the same grid in the eye-tracking experiment. The targets are to visually match the previous plots. Thank you for raising this point.

Reference

Clavagnier, S., Prado, J., Kennedy, H., & Perenin, M.-T. (2007). How humans reach: distinct cortical systems for central and peripheral vision. *The Neuroscientist: A Review Journal Bringing Neurobiology, Neurology and Psychiatry*, *13*(1), 22–27.

González-Alvarez, C., Subramanian, A., & Pardhan, S. (2007). Reaching and grasping with restricted peripheral vision. *Ophthalmic & Physiological Optics: The Journal of the British College of Ophthalmic Opticians*, *27*(3), 265–274.

Kalidindi, H. T., & Crevecoeur, F. (2023). Task dependent coarticulation of movement sequences (p.2023.12.15.571847). https://doi.org/10.1101/2023.12.15.571847

Miller, G. A. (1956). The magical number seven plus or minus two: some limits on our capacity for processing information. *Psychological Review, 63*(2), 81–97.

Sivak, B., & MacKenzie, C. L. (1990). Integration of visual information and motor output in reaching and grasping: the contributions of peripheral and central vision. *Neuropsychologia*, *28*(10), 1095–1116.